DATA RELEASE

# A high-quality draft genome for *Melaleuca alternifolia* (tea tree): a new platform for evolutionary genomics of myrtaceous terpene-rich species

Julia Voelker[1],[*], Mervyn Shepherd[1] and Ramil Mauleon[1]

1 Faculty of Science and Engineering, Southern Cross University, Military Road, East Lismore NSW 2480, Australia

## ABSTRACT

The economically important *Melaleuca alternifolia* (tea tree) is the source of a terpene-rich essential oil with therapeutic and cosmetic uses around the world. Tea tree has been cultivated and bred in Australia since the 1990s. It has been extensively studied for the genetics and biochemistry of terpene biosynthesis. Here, we report a high quality *de novo* genome assembly using Pacific Biosciences and Illumina sequencing. The genome was assembled into 3128 scaffolds with a total length of 362 Mb (N50 = 1.9 Mb), with significantly higher contiguity than a previous assembly (N50 = 8.7 Kb). Using a homology-based, RNA-seq evidence-based and *ab initio* prediction approach, 37,226 protein-coding genes were predicted. Genome assembly and annotation exhibited high completeness scores of 98.1% and 89.4%, respectively. Sequence contiguity was sufficient to reveal extensive gene order conservation and chromosomal rearrangements in alignments with *Eucalyptus grandis* and *Corymbia citriodora* genomes. This new genome advances currently available resources to investigate the genome structure and gene family evolution of *M. alternifolia*. It will enable further comparative genomic studies in Myrtaceae to elucidate the genetic foundations of economically valuable traits in this crop.

**Subjects** Genetics and Genomics, Bioinformatics, Plant Genetics

**Submitted:** 26 April 2021

**\*** Corresponding author. E-mail: j.voelker.10@student.scu.edu.au

Preprint submitted at https://doi.org/10.5281/zenodo.5162127

## DATA DESCRIPTION

### Background

*Melaleuca alternifolia* (Maiden and Betche) Cheel (tea tree) (NCBI:txid164405), is a shrub or small tree that is native to the southern Queensland and northern New South Wales regions of Australia [1] (Figure 1). It belongs to the Myrtaceae, a large angiosperm family of southern hemisphere origin, encompassing 142 genera and over 5500 species [2]. Notably, many Myrtaceae species produce high concentrations of volatile terpenoid compounds, usually stored in schizogenous secretory cavities in their leaves [1]. Of the 17 tribes, the Melelaeucae, Eucalypteae and Myrteae have the highest diversity of unique monoterpenoid and sesquiterpenoid compounds in their leaf oils [3]. These oils are thought to have important adaptive roles, functioning in plant defence against pests and protection against abiotic stresses [4, 5]. The oil is also distilled from the leaves of several species and used in medicinal, therapeutic and cosmetic products [2]. The cultivation of *M. alternifolia* and

**Figure 1.** *Melaleuca alternifolia* (tea tree) is a medium-sized tree with papery bark. (a) A mature tree growing near water. (b) The top of a tree with an abundance of flowers. Pictures courtesy of M. Shepherd.

production of tea tree oil is one of the more important industries based on an essential oil from a myrtaceous species in Australia and overseas [1].

## Context

Here we report a *de novo* high quality draft assembly of the *M. alternifolia* genome, using the short sequencing reads from an earlier draft genome [6] together with newly generated Pacific Biosciences (PacBio) single-molecule real-time (SMRT) sequencing reads. This new draft assembly resulted in a genome size of 362 megabase pairs (Mb), which was close to the size of the previous assembly (356.5 Mb) [6] and the physical size estimated from flow cytometry (357 Mb) [7]. The overall assembly statistics were improved by using longer sequencing reads; the N50 of assembled scaffolds was 1.9 Mb, which is about 214 times higher than the N50 achieved by Calvert *et al.* [6]. The BUSCO (benchmarking universal single-copy orthologs) score for the genome, which assesses the presence of single-copy orthologs [8], was also increased from 86.3% to 98.1%. The subsequent gene annotation led to more than 37,000 genes being predicted, and a high level of ortholog completeness (89.4%) was confirmed for the predicted proteome. Furthermore, the organisation of genes could be resolved in more detail, with many long scaffolds showing synteny to *Eucalyptus grandis* chromosomes.

This new genome sequence for tea tree should meet our aim to generate a resource for comparative studies with other Myrtaceae species to investigate mechanisms of genome evolution in this group. We are especially interested in mechanisms underlying gene family evolution and, in particular, the terpene synthase (*TPS*) gene family. This gene family is responsible for the final stages of terpene biosynthesis and is highly diversified in plants [9]. Tandem duplication is thought to be an important mechanism for gene family evolution in the *TPS* family, and more broadly for adaptive genes in long-lived eucalypts and other tree groups [10]. An earlier analysis of the *TPS* gene family in *M. alternifolia* revealed relatively



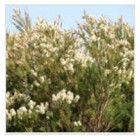

⊚ **Additional information for the creation of a high-quality draft genome for *Melaleuca alternifolia* (tea tree)** ▾

Julia Voelker[1]

[1]Faculty of Science and Engineering, Southern Cross University, Military Road, East Lismore NSW 2480, Australia

dx.doi.org/10.17504/protocols.io.bwi2pcge

👤 Julia Voelker

**Figure 2.** Protocol providing additional information for the creation of a high-quality draft genome for *Melaleuca alternifolia* (tea tree) [12]. https://www.protocols.io/widgets/doi?uri=dx.doi.org/10.17504/protocols.io.bwi2pcge

few *TPS* compared with other myrtaceous species [6]. This new, more contiguous genome will allow more confident exploration of questions on gene family size and microsynteny analysis in tea tree.

## METHODS

### DNA extraction

Fresh, young foliage was collected from the reference genotype SCU1 of *Melaleuca alternifolia*, the same individual used by Calvert *et al.* [6] for Illumina sequencing. To yield high-quality DNA for PacBio sequencing, the cetrimonium bromide (CTAB) extraction protocol from Healey *et al.* [11] was used, with modifications as mentioned in the protocols.io protocol (Figure 2) [12].

The quantity of extracted DNA was measured fluorometrically with a Qubit BR assay (Invitrogen), and the quality was assessed using a Nanodrop (Thermo Fisher Scientific). Furthermore, the integrity of the DNA was examined on a 0.5% agarose gel.

### DNA sequencing

The DNA sample was sent to the Ramaciotti Centre for Genomics (University of New South Wales, Sydney, Australia). Here, further quality control was undertaken using Pippin Pulse gel electrophoresis (Sage Science). DNA was concentrated to 200–250 ng/μl using AMPure PB beads (Pacific Biosciences) so that it was suitable for library preparation, where the aim was to generate DNA fragments of 20–50 kilobase pairs (Kb). The gDNA fragments were sequenced using two PacBio Sequel single-molecule real-time (SMRT) cells. Raw sequencing reads are available from the National Center for Biotechnology Information (NCBI) BioProject PRJNA702189.

### Genome assembly

Before assembly, all reads were mapped to the reference sequence of the *E. grandis* chloroplast (GenBank accession MG925369.1) using minimap2 v2.17-r941 (RRID:SCR_018550) [13]. Reads aligning with more than 80% of their length were filtered out and the remaining reads were used for the nuclear genome assembly.

With the current advances in sequencing technologies, many tools have been developed to meet the demand for long-read assemblers, using different assembly algorithms [14, 15]. However, not all are suited for a diploid, heterozygous plant genome. Hence, three different assembly methods were carried out to compare their performance and select the most suitable tool: (1) Canu v1.8 (RRID:SCR_015880), a long-read assembler [16]; (2) Flye v2.5

(RRID:SCR_017016), a long-read assembler [17]; and (3) MaSuRCA v3.4.0 (RRID:SCR_010691), a hybrid assembler using long-reads and Illumina short-reads [18]. For the MaSuRCA genome assembly, the available short paired-end reads from the same genotype [6] were incorporated into the assembly. Before assembly, BBTools v38.50 (RRID:SCR_016968) [19] was used to filter Illumina reads for sequences of at least 75 bp in length, and duplicate reads were removed using FastUniq (RRID:SCR_000682) [20]. Four libraries were available, with insert sizes of 350 bp, 550 bp, 300 bp, and 700 bp, respectively.

For each assembly, the completeness was assessed using Quast v5.0.2 (RRID:SCR_001228) [21] and BUSCO (Galaxy Version 4.1.2, eudicot-odb10 database, RRID:SCR_015008) [8]. The most suitable assembly tool was selected based on overall assembly length, best BUSCO score and indicators of contiguity, such as N50, NG50, L50, and LG50 values.

Furthermore, the selected assembly was screened for contamination using BlobTools v1.1.1 (RRID:SCR_017618) [22]. The NCBI nucleotide (nt) database [23] was used as reference for BLASTn v2.9.0+ (RRID:SCR_001598) [24], while the UniProt reference proteomes (release May 2020) were input for Diamond blastx v0.9.24 [25]. The required coverage files were created with minimap2 and SAMtools v1.9 (RRID:SCR_002105) [26]. Post contaminant removal, the final genome assembly was deposited at NCBI GenBank with accession number JAGKPW010000000.

The heterozygosity of the genome was calculated by aligning the Illumina paired-end libraries to the final assembly using minimap2. The alignment files were assigned to read groups with Picard v2.23.8 (RRID:SCR_006525) [27], and variants were called using the Genome Analysis Toolkit (GATK v4.1.9.0, RRID:SCR_001876). The heterozygosity ratio was calculated as the number of heterozygous sites versus the total number of nucleotides in the assembly.

## Gene prediction

The Fgenesh++ v7.2.2 pipeline (RRID:SCR_018928) [28] was used to predict genes in the assembled scaffolds. The NCBI non-redundant plant protein database (provided by Softberry), the *Eucalyptus grandis* gene matrix (purchased by the Australian BioCommons), and RNA-seq spliced read alignments were provided as evidence. For the RNA evidence, the *M. alternifolia* RNA-seq data from three other individual trees of the same chemotype (BioProjects PRJNA388506; BioSamples SAMN07178263, SAMN07178261, SAMN07178248) were downloaded and converted to fastq format with the NCBI sequence read archive (SRA) toolkit. They were subjected to quality control using FastQC (RRID:SCR_014583) and trimming with Flexbar (RRID:SCR_013001) [29], following the method by Padovan *et al.* [30]. They were then aligned to the genome using ReadsMap (v1.10.1). Upon completion of the Fgenesh++ run, the predicted genes were filtered to remove incomplete gene models (missing start and/or stop codon) and genes coinciding with repeat regions. Transposable elements in the assembled genome were first identified with RepeatModeler v2.0.1 (RRID:SCR_015027), RepeatMasker v4.1.0 (RRID:SCR_012954) and the Dfam 3.1 database [31]. Genes containing a full-length transposon, or having at least 20% of their sequence overlapping with repeat regions, were determined with Bedtools intersect v2.29.2 (RRID:SCR_006646). Except for sequences containing Pfam domains not related to transposable elements, the identified genes were removed from the prediction. Gene prediction statistics were assessed with BUSCO (proteome mode), InterProScan (Galaxy



version 5.0.0, RRID:SCR_005829) [32] and AGAT v.0.5.1 [33]. With InterProScan, a similarity-based approach was used to screen predicted proteins for sequences listed in the PfamA database (RRID:SCR_004726) [34]. Furthermore, using the taxonomy search of the Pfam database [35], a list of protein families known to be present in eudicotyledons was created to assess which InterProScan results contained those eudicot protein families.

### Synteny with eucalypts and whole genome alignments

The predicted coding sequences (CDS) were aligned to eucalypt reference CDS using CoGe SynMap [36]. Default parameters were retained, allowing a maximum distance of 20 genes between two matches and five genes as the minimum number of aligned pairs. The syntenic path assembly (SPA) was chosen as display option. Reference sequences were the unmasked CDS (v2.0, CoGe id35018) of *E. grandis* strain BRASUZ1 (CoGe id35288; BioProject: PRJNA252394), and the unmasked CDS (v1.1, CoGe id28779) of *Corymbia citriodora* subsp. variegata (CoGe id40461, BioProject: PRJNA234431). The CDS of *Populus trichocarpa* v3 (CoGe id38424 and CoGe:id23993; BioProjects PRJNA17973 and PRJNA10772) was used for comparison.

In addition to the screen for syntenic gene regions, pair-wise whole genome alignments of *M. alternifolia* were also carried out with *E. grandis* and *C. citriodora* pseudo-chromosomes. The genomes were aligned in NUCmer (Nucleotide mummer v3.1, RRID:SCR_018171), with the –mum –minmatch 40 –mincluster 100 options, and matches were visualised using Mummerplot (v3.5) [37].

### Orthogroup discovery

OrthoFinder v2.5.2 (RRID:SCR_017118) [38, 39] was used to discover orthogroups among the protein sequences of selected species and infer phylogenetic relationships. The proteomes of *Arabidopsis lyrata* v2.1 [40], *A. thaliana* ARAPORT11 [41], *E. grandis* v2.0 [42], *P. trichocarpa* v4.1 [43], *Salix purpurea* v1 [44] and *Vitis vinifera* v2.1 [45] were obtained from Phytozome [46], using the primary transcripts only. OrthoFinder was run with default settings to compare the mentioned species with the *C. citriodora* primary transcripts [47] and *M. alternifolia* protein sequences. OrthoFinder first created orthogroup gene trees, which were then used to infer an unrooted species tree with the STAG algorithm [48] and identify the root of the tree using the STRIDE algorithm [49]. The resulting phylogenetic tree was visualised with Dendroscope v3.7.3 [50].

### DATA VALIDATION AND QUALITY CONTROL

### A *de novo* genome assembly for *Melaleuca alternifolia*

The modified DNA extraction protocol yielded high-molecular-weight (HMW) DNA of high purity (260:280 ratio = 1.86, 260:230 ratio = 2.01). As confirmed by Pippin Pulse gel electrophoresis, gDNA was highly intact with a size greater than 40 Kb. Subsequent PacBio sequencing resulted in 1,831,348 reads, with a GC content of 42%, individual sequence lengths of between 50 and 71,437 bp and a total length of 19,732,598,005 bp. Based on a flow cytometry genome size estimate of 356.97 Mb, this implies a sequencing depth of around 55×.

After sequencing, the tea tree genome was assembled with the three different tools Canu, Flye and MaSuRCA. For the MaSuRCA assembly, Illumina sequencing reads [6] were included, which covered the genome more than 300×. Consequently, the Canu and Flye

**Table 1.** Comparison of the different *M. alternifolia de novo* assemblies.

| Assembly | Length (Mb) | No. of scaffolds | N50[a] (Kb) | NG50[b] (Kb) | BUSCO score[c] (%, eudicot_odb10) |
|---|---|---|---|---|---|
| Canu | 451.93 | 3,429 | 864.59 | 1,331.08 | C:97.3 (S:72.7, D:24.6), F:0.8, M:1.9 |
| Flye | 328.17 | 2,197 | 570.96 | 489.41 | C:96.9 (S:93.8, D:3.1), F:0.9, M:2.2 |
| MaSuRCA | 365.23 | 3,217 | 1,882.57 | 1,911.23 | C:98.1 (S:91.2, D:6.9), F:0.6, M:1.3 |
| Illumina draft[d] | 356.50 | 221,396 | 8.78 | 8.75 | C:86.3 (S:84.5, D:1.8), F:6.2, M:7.5 |

[a]N50: the collection of all contigs of that length or longer covering at least half of the genome assembly;
[b]NG50: the collection of all contigs of that length or longer covering at least half of the expected genome size of
357 Mb; [c]C: complete, S: single-copy, D: duplicated, F: fragmented, M: missing; [d]by Calvert *et al.* [6].

assemblies were based on a smaller read dataset of PacBio reads with around 55×
sequencing depth, whereas the combined dataset of PacBio and Illumina reads for the
MaSuRCA assembly resulted in a sequencing depth of around 350×. This difference in
dataset size renders a direct comparison of the three assembly algorithms impossible;
however, it was still decided to compare the assembly statistics of all three tools to make an
informed decision on the best *M. alternifolia* assembly.

Overall, the MaSuRCA assembly had the most favourable assembly and completeness
statistics, in addition to having an assembly size (365 Mb) closest to the flow cytometry
estimate (357 Mb) and the Illumina assembly length of 356.5 Mb [6, 7] (Table 1). The N50 for
the MaSuRCA assembly was 214 times higher than that for the earlier genome assembly by
Calvert *et al.* [6] (N50 = 9 Kb), representing a considerable improvement in scaffold length
owing to the long-read sequencing approach. The N50 of the MaSuRCA assembly
(N50 = 1883 Kb), also exceeded the N50 values for the Canu and Flye assemblies, which
were 865 Kb and 571 Kb, respectively (Table 1).

For each of the three assemblies, BUSCO analysis was conducted to assess their
completeness of orthologous sequences. The MaSuRCA assembly had the highest score
(98.1%) for complete BUSCO groups, followed by Canu (97.3%) and Flye (96.9%) (Table 1).
The MaSuRCA assembly also contained the fewest fragmented and missing reference genes
(1.9%), but the lowest proportion of duplicated orthologs was found in the Flye assembly
(3.1%) (Table 1). Unlike the other assemblies, the Canu assembly had a notably higher
percentage (24%) of complete but duplicated BUSCOs. Taken together with the notably
larger total assembly length for the Canu assembly (451 Mb) relative to the other
approaches, this high percentage of duplicated sequences suggests that Canu might have
assembled more than one haplotype [51]. These findings coincide with recently reported
assembly comparisons, in which Canu resulted in larger than expected assembly sizes
containing uncollapsed haplotypes [52, 53]. In the *M. alternifolia* assembly, this might be
explained by the poor resolution of haplotypes in highly heterozygous regions, as it was
also observed by Guiglielmoni *et al.* [53]. Furthermore, the propensity of PacBio reads to
contain higher error rates [54] might have contributed to this outcome. An increased
sequencing depth followed by the phasing of haplotypes with specific tools should be able
to resolve these errors [16].

It is possible that MaSuRCA exceeded the other two tools owing to the increased
sequencing depth achieved by including Illumina reads. Canu and Flye both reportedly
perform well with a sequencing depth of 50× or less, but their performance is expected to

**Table 2.**  Final assembly statistics for filtered MaSuRCA scaffolds, created with Quast and BUSCO analyses.

| Characteristic | MaSuRCA scaffolds |
|---|---|
| Number of scaffolds | 3128 |
| Largest scaffold, bp | 11,132,794 |
| Total length, bp | 362,036,213 |
| GC content, % | 40.35 |
| N50, bp | 1,894,811 |
| N75, bp | 594,516 |
| L50 | 54 |
| L75 | 135 |
| Number of Ns per 100 Kb | 3.90 |
| Complete and single-copy BUSCOs (%) | 2120 (91.1) |
| Complete and duplicated BUSCOs (%) | 162 (7) |
| Fragmented BUSCOs (%) | 14 (0.6) |
| Missing BUSCOs (%) | 30 (1.3) |
| Total BUSCO groups searched | 2326 (eudictos_odb10) |

improve further with increasing coverage [16, 17]. Particularly for repeat-rich plant genomes, higher PacBio coverage might be required for long-read-only assemblies. Given its higher contiguity and completeness over the other assemblies, MaSuRCA was used for further analyses.

BlobTools was used to screen for contaminating sequences from other species. Sixty contaminated scaffolds were identified in the MaSuRCA assembly (Figure 3). These sequences were classified as bacterial, viral or fungal. However, they only represented 0.3% of the total assembly length, and most of the assembly (353.6 Mb) was identified as plant sequences. Based on this screening, contaminated scaffolds were removed, while all scaffolds listed as Streptophyta, or those with no hits to the databases were retained. Of the raw sequencing reads used in the assembly, 97.68% mapped to the scaffolds, and only 0.03% mapped to contaminated sequences. A further contaminant screen upon submission of the assembly to the NCBI GenBank portal led to the exclusion of 29 scaffolds of probable mitochondrial or chloroplast origin.

After removing identified contaminants, new assembly statistics were created in Quast (Table 2). BUSCO analysis on the filtered genome assembly showed a high level of completeness with 2282 (98.1%) of the 2326 BUSCO groups being reported as complete (Table 2). Most (91.1%) were single-copy orthologs. Variant calling using Illumina reads aligned to the assembled sequence revealed a heterozygosity rate of 0.6% for this *M. alternifolia* individual.

## Fgenesh++ prediction results

A total of 52,135 protein-coding genes were predicted by Fgenesh++, of which 23,920 were predicted based on protein evidence (high sequence similarity to NCBI nr-protein plant database), with the remaining genes being predicted *de novo*. A total of 2170 genes had missing start or stop codons. A further 12,739 genes were excluded because of overlaps with transposable elements, resulting in a filtered set of 37,226 complete gene models.

BUSCO assessment of the translated sequences revealed an overall high level of ortholog completeness in the proteome after filtering the predictions (89.4%), with most of them being single-copy (Table 3). Furthermore, a screen against the reference database of PfamA protein families showed that 64.39% of all proteins contain known eudicot Pfam domains.

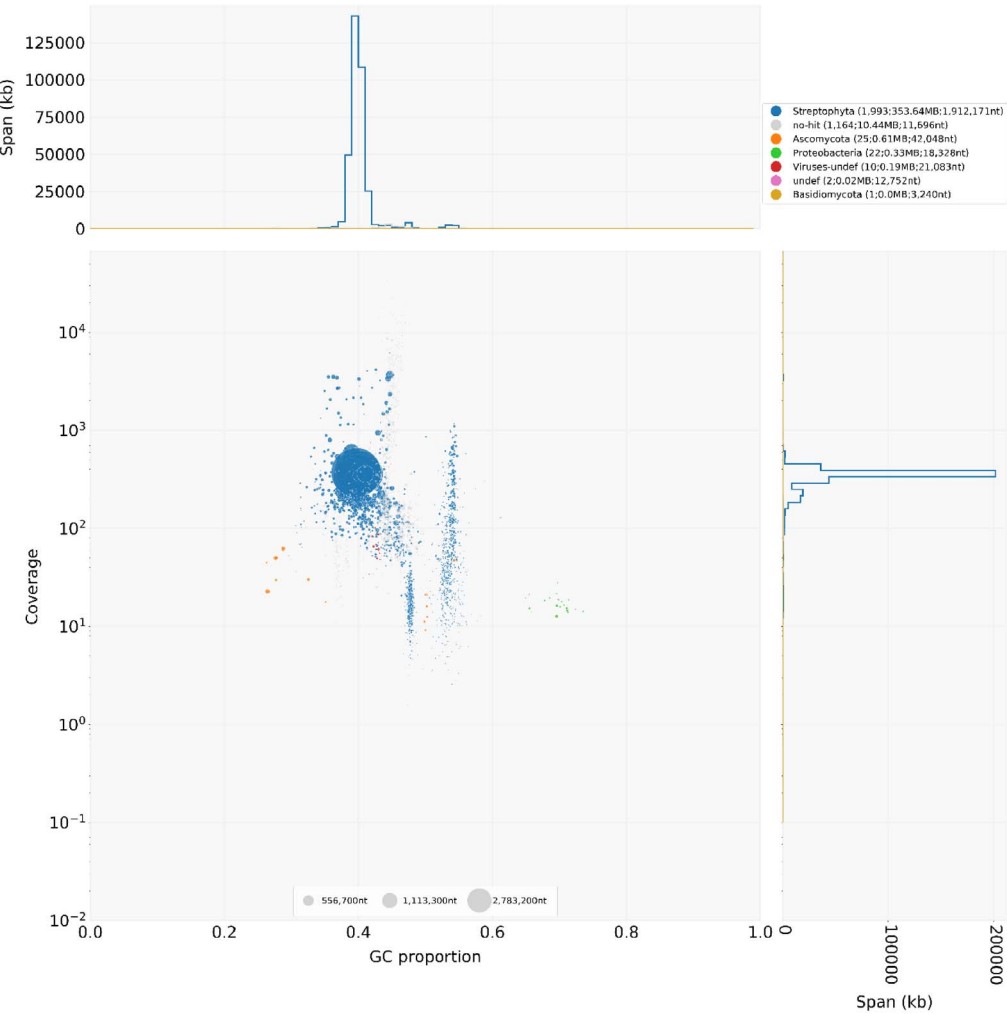

**Figure 3.** BlobPlot showing potential contaminations in the assembled MaSuRCA scaffolds. Each dot represents a single scaffold, and the larger the dot, the longer the represented scaffold sequence. Scaffolds were assigned to taxonomic groups based on their alignment to the NCBI nucleotide database and the UniProt protein database. The scaffold coverage is based on alignments of the raw sequencing reads to the assembled scaffolds.

**Table 3.** Fgenesh++ filtered gene prediction results.

| Trait | Count |
|---|---|
| Number of predicted complete gene models | 37,226 |
| Number of proteins with hit to Pfam domains | 26,415 |
| Number of proteins with hit to eudicot Pfam domains | 23,970 |
| Percentage of genes with eudicot Pfam domains, % | 64.39 |
| Mean gene length, bp | 2,791 |
| Complete and single-copy BUSCOs (%) | 1938 (83.3) |
| Complete and duplicated BUSCOs (%) | 141 (6.1) |
| Fragmented BUSCOs (%) | 114 (4.9) |
| Missing BUSCOs (%) | 133 (5.7) |
| Total BUSCO groups searched (eudicots_odb10) | 2326 |

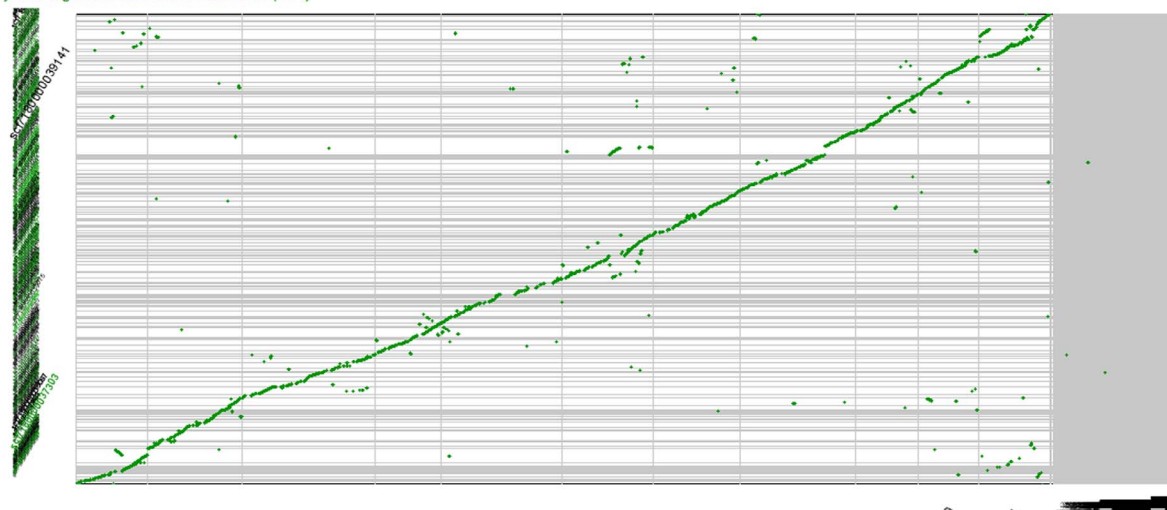

*y-axis organism: Melaleuca alternifolia (v1.2)*

*x-axis organism: Eucalyptus grandis strain BRASUZ1 (v2.0)*

**Figure 4.** Syntenic map (SynMap) of the predicted coding sequences of *M. alternifolia* and *E. grandis*. The syntenic path assembly shows the *M. alternifolia* scaffolds (*y*-axis) ordered according to their synteny to the *E. grandis* chromosomes (*x*-axis), which were ordered numerically. Each green dot represents a syntenic gene region. The *E. grandis* scaffolds that are not part of the chromosome-scale assembly are displayed at the end of the *x*-axis (grey area).

## Synteny to eucalypts

The predicted coding sequences (CDS) of *M. alternifolia* were aligned to eucalyptus reference CDS using CoGe Synmap. The resulting syntenic map showed a high level of collinearity in the gene order of *M. alternifolia* and *E. grandis* (Figure 4). Since the syntenic path assembly ordered scaffolds based on their synteny to the reference genome of *E. grandis*, no conclusions can be drawn about the orientation of short scaffolds containing too few genes. However, the assembled scaffolds of sufficient size reveal rearrangements on the chromosome scale; some inversions in the tea tree genome were evident (*E. grandis* chromosome one), together with one translocation between the *E. grandis* chromosomes eight and six. Comparison with *C. citriodora* CDS showed a similar plot, but there were more gaps, especially on *C. citriodora* chromosomes 4 and 5 (see syntenic maps in GigaDB [55]). Furthermore, a comparison of *M. alternifolia* with *P. trichocarpa* revealed that synteny is still identifiable between these two rosids. Nevertheless, the synteny was reduced, with larger genetic distance leading to translocations and inversions being more notable than among the examined Myrtaceae.

Whole-genome alignments with Mummerplot resulted in similar findings, with an overall high sequence similarity among the compared Myrtaceae, with more inversions in the tea tree alignment to *C. citriodora* than to *E. grandis* (see whole-genome alignments Mummerplot in GigaDB [55]). The underlying cause of the different densities in the pairwise synplots between *Melaleuca, Corymbia* and *Eucalyptus*, and differences in the rearrangements detected between the pairs will require more investigation.

*M. alternifolia* is an ideal model species to compare genomic features with eucalypts. The two tribes Melaleuceae and Eucalypteae share some anatomical features, such as capsular fruit, an abundance of oil glands in their leaves, and epicormic buds underneath their bark [1, 2]. Their sequence similarities are expected to be high, with genetic markers being



transferable to some extent [56]. The syntenic comparisons and whole-genome alignments of our study confirmed these expectations.

## Orthogroup discovery

The inference of phylogenetic relationships and the discovery of orthogroups based on the sequences of selected species affirmed the fgenesh++ gene prediction results for *M. alternifolia.*

OrthoFinder assigned 248,528 genes (91.3% of total) to 29,672 orthogroups. Of these, 10,982 orthogroups contained genes from all eight species, while 4386 groups were species-specific. Most of the *M. alternifolia* genes (78%) were assigned to orthogroups with two or more species present (see tables in GigaDB for more OrthoFinder-related statistics [55]).

Phylogenetic relationships were inferred based on the gene trees created by OrthoFinder for the eight selected species [37]. The resulting species tree showed *Arabidopsis* as an outgroup, while the Myrtaceae members *E. grandis, C. citriodora* and *M. alternifolia* form a clade that separated from the other rosids *P. trichocarpa, S. purpurea* and *V. vinifera* (Figure 5). Further investigations of the *M. alternifolia* sequences should be conducted to understand the dynamics of gene family evolution in this species, especially compared with other Myrtaceae.

The tea tree genome will make a valuable addition to the growing quiver of sequences available for Myrtaceae. So far, genomic studies on Myrtaceae have mainly been motivated by the economic importance of some species and the relevance to identify the genetic foundations of specific traits [57]. Hence, the genomes of *E. grandis* and *E. globulus* were the first trees of the Myrtaceae family to be sequenced [42], but these were soon followed by the genome of a sister genus, *C. citriodora* [47, 58]. Their geographical isolation, thus their separated evolution in Australia, make eucalypts a model taxon for comparative genomic studies to find shared evolutionary history, but also to find unique genome characteristics, compared with other woody perennials of the rosid lineage, such as *Populus* and *V. vinifera* [42].

In addition to providing an outgroup for phylogenetic analyses of derived and ancestral characters in the eucalypts, the *M. alternifolia* genome resource will help to elucidate genomic responses to adaptive pressures among the Myrtaceae evolving on the Australian continent.

Although the evolutionary history of the *Melaleuca* remains uncertain, the genera *Eucalyptus* and *Melaleuca* are thought to have had a shared ancestry until 68 mega annum (Ma) [59], and current evidence suggests that *Melaleuca* and *Eucalyptus* largely adapted to contrasting environments [60]. Most of the genus *Melaleuca* are small trees or shrubs that grow in wetland or periodically waterlogged habitats [1], and have developed adaptations consistent with their waterlogged life history. For example, *Melaleuca* trees are capable of gas exchange through their bark [61], and seedlings develop aquatic roots as well as changed leaf morphology when submerged [62, 63]. Thus, melaleucas are well-adapted to permanently or seasonally wet habitats. In contrast, most eucalypts are adapted to survive in arid environments with frequently occurring fires [64]. This divergence in their habitat might have evoked distinct adaptive responses in eucalypts and melaleucas.

The genome of another related species, *Leptospermum scoparium* (mānuka), has recently been sequenced [65]. Phylogenetic studies on Myrtaceae indicate that *Leptospermum* and



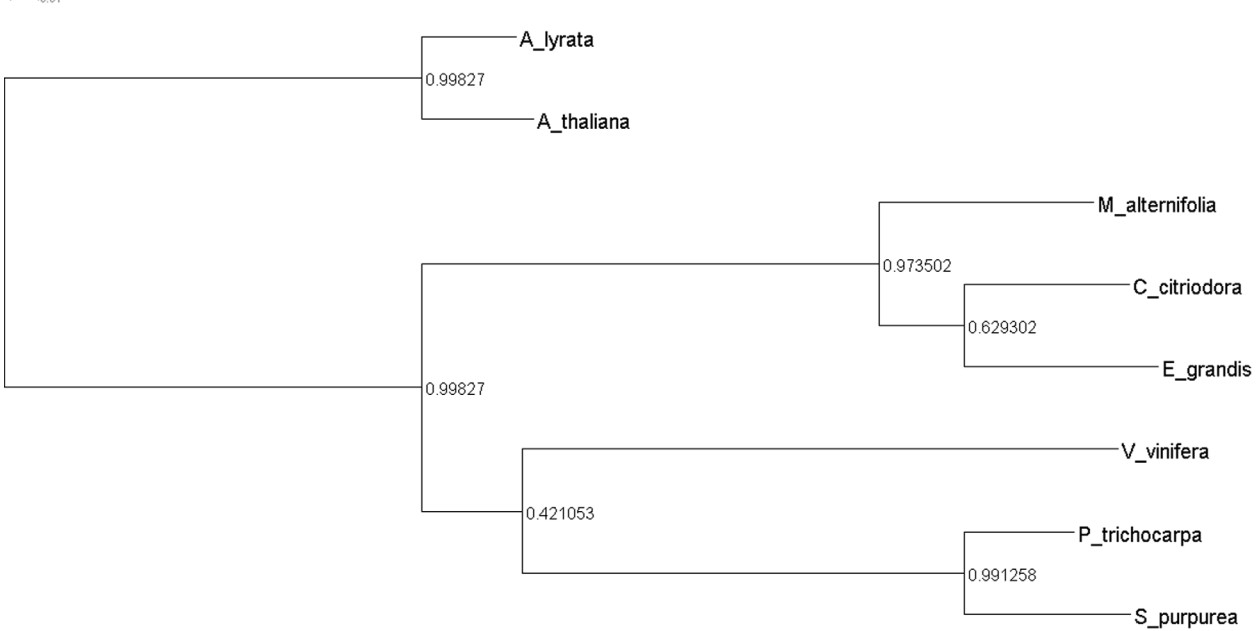

**Figure 5.** Species tree as inferred by OrthoFinder with branch lengths based on substitutions per site.

*Eucalyptus* diverged 62 Ma [47, 66]. Thrimawithana *et al.* [65] reported high levels of overall synteny between *E. grandis* and *L. scoparium*, with most orthologous regions being located on the same chromosome in both species. Our analysis of the *M. alternifolia* genome also showed a high degree of shared synteny with *E. grandis* and *C. citriodora*. This shared synteny at the scaffold level, together with the great number of orthologous genes among these three species, suggests that this newly assembled *M. alternifolia* genome is an excellent resource to increase our understanding of gene clustering and the evolution of tandemly duplicated genes in Myrtaceae. Further comparative genomic studies between the genomes of *M. alternifolia, L. scoparium* and all published eucalypt genomes [42, 47, 58, 67] could help to elucidate the genomic mechanisms underlying the adaptive evolution of different Myrtaceae and reveal lineage-specific gene family expansions. Families of interest include *TPS* genes and other genes involved in stress responses. In *E. grandis*, for example, most resistance (R) genes containing nucleotide-binding sites and leucine-rich repeat domains (NBS-LRR) were shown to be organised in tandem clusters [68], as well as genes encoding for S-domain receptor-like kinases (SDRLK) and MYB transcription factors [42, 69]. These findings indicate that tandem duplication is essential for adaptation to dynamic environments, especially for genes involved in responses to abiotic and biotic stresses. Therefore, comparison of how similarly these gene families evolved in Myrtaceae, and which selection pressures might have influenced their genome structure, is warranted.

## REUSE POTENTIAL

*De novo* assembly of the *M. alternifolia* genome using MaSuRCA with PacBio long-reads and Illumina short-reads resulted in a high-quality draft genome that was similar in length (362 Mb) to the previously reported short-read assembly (357 Mb) [6], but with considerably longer scaffolds. The N50 value was increased by a factor of 214. The completeness of the



hybrid assembly was also improved, as indicated by 6.1% fewer BUSCOs missing from the assembly, and a decrease of 5.6% fewer fragmented BUSCOs than the previous draft. Alignment of *M. alternifolia* sequences to *E. grandis* and *C. citriodora* indicated high sequence similarities and correlations in gene order among the three Myrtaceae. The longer scaffolds of this new assembly will help to illuminate genome organisation in *M. alternifolia* and will allow further exploration of the significance of tandem gene duplication as a mechanism of gene family evolution in tea tree and related Myrtaceae species.

## DATA AVAILABILITY

This whole genome sequencing project has been deposited at NCBI GenBank under the accession JAGKPW000000000. The assembly version described in this paper is version JAGKPW010000000. PacBio raw sequencing reads are available under BioProject PRJNA702189. Supplementary information regarding the computational methods has been summarised at protocols.io [12]. Other supporting data are available in the *GigaScience* GigaDB repository [55].

## DECLARATIONS
## LIST OF ABBREVIATIONS

bp: base pairs; BUSCO: Benchmarking Universal Single-Copy Orthologs; CDS: coding sequence; Kb: kilobase pairs; Ma: mega annum; Mb: megabase pairs; NCBI: National Center for Biotechnology Information; PacBio: Pacific Biosciences

## ETHICAL APPROVAL

Not applicable.

## CONSENT FOR PUBLICATION

Not applicable.

## COMPETING INTERESTS

The authors declare that they have no competing interests.

## FUNDING

This project was funded by Southern Cross University and the Australian Tea Tree Industry Association (ATTIA).

## AUTHORS' CONTRIBUTIONS

Formal analysis: J.V.; Funding acquisition: M.S.; Investigation: J.V.; Design of methodology: J.V., M.S., R.M.; Writing – original draft: J.V.; Writing – review and editing: J.V., M.S., R.M.; Supervision – M.S., R.M. All authors read and approved the final version of this manuscript.

## ACKNOWLEDGEMENTS

The Fgenesh++ license and required computing resources for gene prediction were provided by the Australian BioCommons and the Pawsey Supercomputing Centre. BUSCO and InterProScan analyses were run on the public servers of Galaxy Australia [70] and Galaxy Europe [71], respectively.



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
