## [Reviewer Report]

Comments on revised manuscriptThe authors made the reasonable answer and added an additional information to the GigaDB repository respectively. It is feasible to publish it according to my personal opinion.

---

## [Reviewer Report]

Comments on revised manuscriptComments to DRR-202104-02-R01

The authors have addressed most of my concerns, but there are still a few grammatical mistakes. 

Line 31: “an essential oil” should be “essential oil”.
Line 35: “high quality” should be “high-quality”.
Line 45: “for the tea tree”.
Line 129: “as a display potion”.
Line 241: “a larger genetic distance”.
Line 259: “Table 1 and 2”.
Line 261: “phylogenetic relationships” and “as an outgroup”.

---

## [Reviewer Report]

Reviewer name and names of any other individual's who aided in reviewer Masaomi HatakeyamaDo you understand and agree to our policy of having open and named reviews, and having your review included with the published papers. (If no, please inform the editor that you cannot review this manuscript.)YesIs the language of sufficient quality?YesPlease add additional comments on language quality to clarify if needed
Are all data available and do they match the descriptions in the paper? YesAdditional CommentsAre the data and metadata consistent with relevant minimum information or reporting standards? See GigaDB checklists for examples <a href="http://gigadb.org/site/guide" target="_blank">http://gigadb.org/site/guide</a>YesAdditional CommentsIs the data acquisition clear, complete and methodologically sound?YesAdditional CommentsIs there sufficient detail in the methods and data-processing steps to allow reproduction?YesAdditional CommentsThe authors wrote, in line 306, Supplementary information regarding the computational methods has been summarised at protocols.io.

However, I cannot access it for now. The method description is enough in natural language, but it would be difficult to reproduce the results only by such a description, and I expect that all the necessary information will be uploaded for reproducibility such as the executed environmental information, command arguments, options, and parameters and so on.
Is there sufficient data validation and statistical analyses of data quality? YesAdditional CommentsIs the validation suitable for this type of data?YesAdditional CommentsIs there sufficient information for others to reuse this dataset or integrate it with other data?YesAdditional CommentsAny Additional Overall Comments to the Author In this manuscript, three types of methods for the de novo genome assembly on Melaleuca alternifolia (tea tree) are evaluated. The authors conducted the following three de novo assemblers, 1) Canu, 2) Flye, and 3) MaSuRCA. Only with MaSuRCA, the Illumina short reads which have been published are used together with the newly sequenced PacBio long reads data. Gene annotation was done on the three assemblies by Fgenesh++ in the same manner and conditions. The assemblies and annotations were evaluated with several criteria such as N50 (continuity), BUSCO (single-copy orthologs validation). Then, by using the MaSuRCA result (assembly), comparative genomics analysis was conducted.

The assemblies and annotations are overall well explained and examined, and it is worth publishing the assembled genome for the scientific and industrial communities. However, I found some typos and unclear sentences. Please check and correct the following points:

* 100 (RRID:SCR_006525) [25], and variants were called using The Genome Analysis Toolkit

This must be a typo, "The" should be small letters or it is not needed here.

* 151 (. Based on a flow cytometry genome size estimate

It is a typo. The period should be out of the brackets, or maybe the bracket should be removed.

Please check the following Errors and whether reference sources are correct

* 181 scaffolds in the MaSuRCA assembly (Error! Reference source not found.).
* 215 of M. alternifolia and E. grandis (Error! Reference source not found.).
* 244 other? rosids P. trichocarpa, S. purpurea and V. vinifera (Error! Reference source not found.).


* 242 The resulting species tree showed Arabidopsis as outgroup, while the Myrtaceae members E. grandis, C. citriodora and M. alternifolia form a clade that separated from the other? rosids P. trichocarpa, S. purpurea and V. vinifera

This sentence is incorrect grammatically. It may be related to this part, and there is no sentence found to refer to Figure 3. Additionally, it would be worth describing which method (e.g. Neighbor-joining, UPGMA, etc.) is used for inferring the phylogenetic tree.

RecommendationAccept

---

## [Reviewer Report]

Reviewer name and names of any other individual's who aided in reviewer Li HLDo you understand and agree to our policy of having open and named reviews, and having your review included with the published papers. (If no, please inform the editor that you cannot review this manuscript.)YesIs the language of sufficient quality?YesPlease add additional comments on language quality to clarify if needed
Are all data available and do they match the descriptions in the paper? YesAdditional CommentsAre the data and metadata consistent with relevant minimum information or reporting standards? See GigaDB checklists for examples <a href="http://gigadb.org/site/guide" target="_blank">http://gigadb.org/site/guide</a>YesAdditional CommentsIs the data acquisition clear, complete and methodologically sound?YesAdditional CommentsIs there sufficient detail in the methods and data-processing steps to allow reproduction?YesAdditional CommentsIs there sufficient data validation and statistical analyses of data quality? NoAdditional CommentsIs the validation suitable for this type of data?YesAdditional CommentsIs there sufficient information for others to reuse this dataset or integrate it with other data?YesAdditional CommentsAny Additional Overall Comments to the AuthorVoelker and his colleagues reported a high-quality de novo genome assembly using PacBio and Illumina sequencing data as a platform for comparative genomics in the Myrtaceae. The concept and part of results sound attractive, however, some details should be inquired. Following are my major concerns about this work.

1. The PacBio sequencing implies a genome coverage (sequencing depth) of around 55x. Why the authors sequenced such a coverage? 80x or more would be a good improvement.

2. The authors assembled the genome using PacBio and Illumina sequencing data. However some important information of the results are not clear. Such as, what about the percentage of repeat sequences and heterozygosity of the tea tree genome in the latest version ?

3. Three different tools Canu, Flye and MaSuRCA were used to assemble the tea tree genome, with a larger total assembly length for the Canu assembly (451 Mb), whereas, below 365.23 Mb generated by other tools. It is difficult to understand, how could such a difference exist. Why Canu could might have assembled more than one haplotype?
RecommendationMajor Revision

---

## [Reviewer Report]

Reviewer name and names of any other individual's who aided in reviewer Yue ZhangDo you understand and agree to our policy of having open and named reviews, and having your review included with the published papers. (If no, please inform the editor that you cannot review this manuscript.)YesIs the language of sufficient quality?YesPlease add additional comments on language quality to clarify if needed
Are all data available and do they match the descriptions in the paper? YesAdditional CommentsAre the data and metadata consistent with relevant minimum information or reporting standards? See GigaDB checklists for examples <a href="http://gigadb.org/site/guide" target="_blank">http://gigadb.org/site/guide</a>YesAdditional CommentsIs the data acquisition clear, complete and methodologically sound?YesAdditional CommentsIs there sufficient detail in the methods and data-processing steps to allow reproduction?YesAdditional CommentsIs there sufficient data validation and statistical analyses of data quality? YesAdditional CommentsIs the validation suitable for this type of data?YesAdditional CommentsIs there sufficient information for others to reuse this dataset or integrate it with other data?YesAdditional CommentsAny Additional Overall Comments to the AuthorComments to DRR-202104-02
This manuscript reported an updated genome database for M. alternifolia, which was mainly used for investigating the genome evolution in Myrtaceae. I think this database will be useful for the community. Even though, I have some suggestions that may help to improve this database and manuscript.

My first issue with this manuscript was the three assemblies by using different methods and datasets. The Canu and Flye assemblies only used the SMRT dataset, however, a bigger dataset including previous Illumina sequencing reads and SMRT reads was used for MaSuRCA assembly. And the results showed that the MaSuRCA assembly was better than the other two assemblies. With different sizes of datasets, we cloud not compare the performance for different genome assembly algorithms. The bigger dataset means more sequencing depth, which reasonably results in better genome assembly. If this paper wants to compare the algorithms, the dataset should be the same. 

By using the Fgenesh++ pipeline, a total of 37,226 gene models with a complete open reading frame (ORF) of the reference genome were predicted. The genes without ORF or overlapping with transposable elements were filtered out, therefore, the number of filtered genes in the main Table3 makes little sense. Besides, I suggested that the authors mapped the gene sequencing to different databases, such as Gene ontology (GO) and Kyoto Encyclopedia of Genes and Genomes (KEGG), to complete the gene functional annotation. 

Only the M. alternifola CDS were mapped to the E. grandis genome for identifying the gene order change. I wonder to know how the authors deal with the small scaffolds that might contain single or few genes. Besides, what biological functions of the inversion or translocation genes are? In the introduction, a previous study suggested the Melelaeucae and Eucalypteae have a diversity of unique mono- and sesquiterpenoid compounds in their leaf oils. Whether the genes with changed order resulted from genome structure variation and these genes, especially in TPS gene family, contributed to the diversity of sesquiterpenoid compounds need further analysis and discussion.

Small issues:
Line 151. The definition of genome coverage is different from the sequencing depth, please confirm whether both genome coverage and sequencing depth are around 55x.

Line 181. Which color dots means the contaminated scaffolds from bacterial, viral, or fungal, there is no explains in the figure legend or manuscript.

Line 275. “suggest” should be “suggests”.


RecommendationMajor Revision